# Guiding Policies with Language via Meta-Learning

**John D. Co-Reyes**[*] **Abhishek Gupta  Suvansh Sanjeev  Nick Altieri  Jacob Andreas**
**John DeNero  Pieter Abbeel  Sergey Levine**
University of California, Berkeley

## Abstract

Behavioral skills or policies for autonomous agents are conventionally learned from reward functions, via reinforcement learning, or from demonstrations, via imitation learning. However, both modes of task specification have their disadvantages: reward functions require manual engineering, while demonstrations require a human expert to be able to actually perform the task in order to generate the demonstration. Instruction following from natural language instructions provides an appealing alternative: in the same way that we can specify goals to other humans simply by speaking or writing, we would like to be able to specify tasks for our machines. However, a single instruction may be insufficient to fully communicate our intent or, even if it is, may be insufficient for an autonomous agent to actually understand how to perform the desired task. In this work, we propose an interactive formulation of the task specification problem, where iterative language corrections are provided to an autonomous agent, guiding it in acquiring the desired skill. Our proposed language-guided policy learning algorithm can integrate an instruction and a sequence of corrections to acquire new skills very quickly. In our experiments, we show that this method can enable a policy to follow instructions and corrections for simulated navigation and manipulation tasks, substantially outperforming direct, non-interactive instruction following.

## 1 Introduction

Behavioral skills or policies for autonomous agents are typically specified in terms of reward functions (in the case of reinforcement learning) or demonstrations (in the case of imitation learning). However, both reward functions and demonstrations have downsides as mechanisms for communicating goals. Reward functions must be engineered manually, which can be challenging in real-world environments, especially when the learned policies operate directly on raw sensory perception. Sometimes, simply defining the goal of the task requires engineering the very perception system that end-to-end deep learning is supposed to acquire. Demonstrations sidestep this challenge, but require a human demonstrator to actually be able to perform the task, which can be cumbersome or even impossible. When humans must communicate goals to each other, we often use language. Considerable research has also focused on building autonomous agents that can follow instructions provided via language (Branavan et al. (2009); Chen & Mooney (2011)). However, a single instruction may be insufficient to fully communicate the full intent of a desired behavior. For example, if we would like a robot to position an object on a table in a particular place, we might find it easier to guide it by telling it which way to move, rather than verbally defining a coordinate in space. Furthermore, an autonomous agent might be unable to deduce *how* to perform a task from a single instruction, even if it is very precise. In both cases, interactive and iterative corrections can help resolve confusion and ambiguity, and indeed humans often employ corrections when communicating task goals to each other.

In this paper, our goal is to enable an autonomous agent to accept instructions and then iteratively adjust its policy by incorporating interactive *corrections* (illustrated in Figure 1). This type of in-the-loop supervision can guide the learner out of local optima, provide fine-grained task definition, and is natural for humans to provide to the agent. As we discuss in Section 2, iterative language corrections can be substantially more informative than simpler forms of supervision, such as preferences, while being substantially easier and more natural to provide than reward functions or demonstrations.

---

[*]jcoreyes@eecs.berkeley.edu

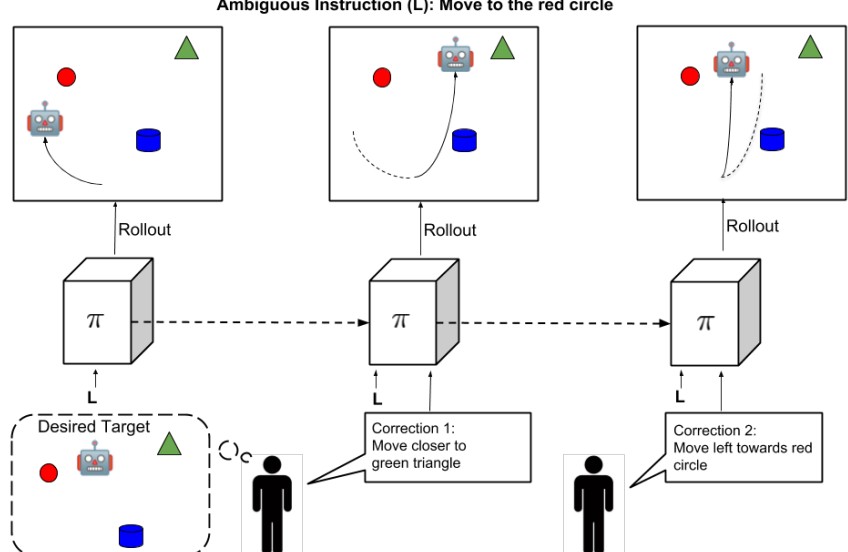

Figure 1: An example where corrections disambiguate an instruction. The agent is unable to fully deduce the user's intent from the instruction alone and iterative language corrections guide the agent to the correct position. Our method is concerned with meta-learning policies that can ground language corrections in their environment and use them to improve through iterative feedback.

In order to effectively use language corrections, the agent must be able to ground these corrections to concrete behavioral patterns. We propose an end-to-end algorithm for grounding iterative language corrections by using a multi-task setup to *meta-train* a model that can ingest its own past behavior and a correction, and then correct its behavior to produce better actions. During a meta-training phase, this model is iteratively retrained on its own behavior (and the corresponding correction) on a wide distribution of known tasks. The model learns to correct the types of mistakes that it actually tends to make in the world, by interpreting the language input. At meta-test time, this model can then generalize to new tasks, and learn those tasks quickly through iterative language corrections.

The main contributions of our work are the formulation of guided policies with language (GPL) via meta-learning, as well as a practical GPL meta-learning algorithm and model. We train on English sentences sampled from a hand-designed grammar as a first step towards real human-in-the-loop supervision. We evaluate our approach on two simulated tasks - multi-room object manipulation and robotic object relocation. The first domain involves navigating a complex world with partial observation, seeking out objects and delivering them to user-specified locations. The second domain involves controlling a robotic gripper in a continuous state and action space to move objects to precise locations in relation to other objects. This requires the policy to ground the corrections in terms of objects and places, and to control and correct complex behavior.

## 2 RELATED WORK

Tasks for autonomous agents are most commonly specified by means of reward functions (Sutton & Barto, 1998) or demonstrations (Argall et al., 2009). Prior work has studied a wide range of different techniques for both imitation learning (Ziebart et al., 2008; Abbeel & Ng, 2004) and reward specification, including methods that combine the two to extract reward functions and goals from user examples (Fu et al., 2018; Thomaz et al., 2006) and demonstrations (Fu et al., 2017; Liu et al., 2017). Other works have proposed modalities such as preferences (Christiano et al., 2017) or numerical scores (Warnell et al., 2017). Natural language presents a particularly appealing modality for task specification, since it enables humans to communicate task goals quickly and easily. Unlike demonstrations, language commands do not require being able to perform the task. Unlike reward functions, language commands do not require any manual engineering. Finally, in comparison to low-bandwidth supervision modalities, such as examples of successful outcomes or preferences,

language commands can communicate substantially more information, both about the goals of the task and how it should be performed.

A considerable body of work has sought to ground natural language commands in meaningful behaviors. These works typically use a large supervised corpus in order to learn policies that are conditioned on natural language commands (MacMahon et al., 2006; Branavan et al., 2009; Vogel & Jurafsky, 2010; Chen & Mooney, 2011; Tellex et al., 2011; Artzi & Zettlemoyer, 2013; Kim & Mooney, 2013; Andreas & Klein, 2015; Misra et al., 2017a; Andreas et al., 2018; Oh et al., 2017). Other works consider using a known reward function in order to learn how to ground language into expert behaviors (Janner et al., 2018; Andreas & Klein, 2015). Most of these works consider the case of instruction following. However, tasks can often be quite difficult to specify with a single language description, and may require interactive guidance in order to be achieved. We focus on this setting in our work, where the agent improves its behavior via iterative language corrections.

While the focus in our work is on incorporating language corrections, several prior works have also studied reinforcement learning and related problems with in-the-loop feedback of other forms Akrour et al. (2011); Pilarski et al. (2011); Akrour et al. (2012); El Asri et al. (2016); Wang et al. (2016b); Warnell et al. (2017); Reddy et al. (2018). In contrast, we study how to incorporate language corrections, which are more natural for humans to specify and can carry more information about the task. However, language corrections also present the challenge that the agent must learn how to ground them in behavior. To this end, we introduce an end-to-end algorithm that directly associates language with changes in behavior without intermediate supervision about object identities or word definitions.

Our approach to learning to learn from language corrections is based on meta-reinforcement learning. In meta-reinforcement learning, a meta-training procedure is used to learn a procedure (represented by initial network weights or a model that directly ingests past experience) (Schmidhuber, 1987) that can adapt to new tasks at meta-test time. However, while prior work has proposed meta-reinforcement learning for model-free RL (Wang et al., 2016a; Duan et al., 2016; Finn et al., 2017; Mishra et al., 2017), model-based RL (Clavera et al., 2018), a wide range of supervised tasks (Snell et al., 2017; Santoro et al., 2016; Vinyals et al., 2016; Sung et al., 2017), as well as goal specification (Xu et al., 2018; Xie et al., 2018), to our knowledge no prior work has proposed meta-training of policies that can acquire new tasks from iterative language corrections.

## 3  PROBLEM FORMULATION

We consider the sequential decision making framework, where an agent observes states $s \in \mathcal{S}$, chooses to execute actions $a \in \mathbf{A}$ and transitions to a new state $s'$ via the transition dynamics $\mathcal{T}(s'|s,a)$. For goal directed agents, the objective is typically to learn a policy $\pi_\theta$ that chooses actions enabling the agent to achieve the desired goal. In this work, the agent's goal is specified by a language instruction $L$. This instruction describes what the general objective of the task is, but may be insufficient to fully communicate the intent of a desired behavior.

The agent can attempt the task multiple times, and after each attempt, the agent is provided with a language *correction*. Each attempt results in a trajectory $\tau = (s_0, a_0, s_1, a_1, ...., s_T, a_T)$, the result of the agent executing its policy $\pi_\theta(a|s, L)$ in the environment. After each attempt, the user generates a correction according to some unknown stochastic function of the trajectory $c \sim \mathcal{F}(\tau)$. Here, cis a language phrase that indicates how to improve the current trajectory $\tau$ to bring it closer to accomplishing the goal. This process is repeated for multiple trials, and we will use $\tau_i$ to denote the trajectory on the $i^{\text{th}}$ trial, and $c_i$ to denote the corresponding correction. An effective model should be able to incorporate these corrections to come closer to achieving the goal. This process is illustrated in Figure 1.

In the next section, we will describe a model that can incorporate iterative corrections, and then describe a meta-training procedure that can train this model to incorporate corrections effectively.

## 4  THE LANGUAGE-GUIDED POLICY LEARNING MODEL

As described in Section 3, our model for guiding policies with language (GPL) must take in an initial language instruction, and then iteratively incorporate corrections after each attempt at the task. This

requires the model to ground the contents of the correction in the environment, and also interpret it in the context of its own previous trajectory so as to decide which actions to attempt next. To that end, we propose a deep neural network model, shown in Figure 2, that can accept the instruction, correction, previous trajectory, and state as input. The model consists of three modules: an instruction following module, a correction module, and a policy module.

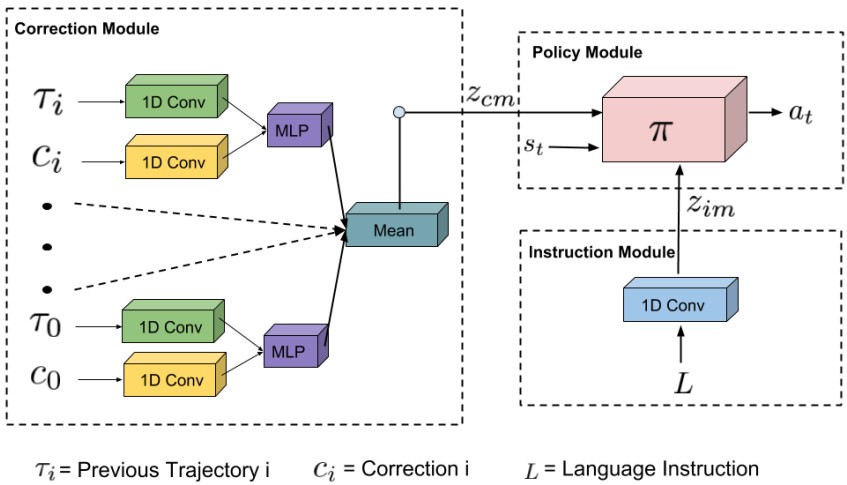

$\tau_i$ = Previous Trajectory i    $c_i$ = Correction i    $L$ = Language Instruction

Figure 2: The architecture of our model. The instruction module embeds the initial instruction $L$, while the correction modules embed the trajectory $\tau_i$ and correction $c_i$ from each previous trial. The features from these corrections are pooled and provided to the policy, together with the current state $s$ and the embedded initial instruction.

The instruction following module interprets the initial language instruction. The instructions are provided as a sequence of words which is converted into a sequence of word-embeddings. A 1D CNN processes this sequence to generate an instruction embedding vector $z_{im}$ which is fed into the policy module.

The correction module interprets the previous language corrections $\mathbf{c}_n = (c_0, \ldots, c_n)$ in the context of the previous trajectories $\boldsymbol{\tau}_n = (\tau_0 \ldots, \tau_n)$. Each previous trajectory is processed by a 1D CNN to generate a trajectory embedding. The correction $c_i$, similar to the language description, is converted into a sequence of word-embeddings which is then fed through a 1D CNN. The correction and trajectory embeddings are then concatenated and transformed by a MLP to form a single tensor. These tensors are then averaged to compute the full correction history tensor $z_{cm} = \frac{1}{n} \sum_{j=0}^{n} \text{MLP}(1\text{dCNN}(\tau_j), 1\text{dCNN}(c_j))$.

The policy module has to integrate the high level description embedded into $z_{im}$, the actionable changes from the correction module $z_{cm}$, and the environment state $s$, to generate the right action. This module inputs $z_{cm}$, $z_{im}$ and $s$ and generates an action distribution $p(a|s)$ that determine how the agent should act. Specific architecture details are described in the appendix.

Note that, by iteratively incorporating language corrections, such a model in effect implements a *learning algorithm*, analogously to meta-reinforcement learning recurrent models proposed in prior work that read in previous trajectories and rewards (Duan et al., 2016; Wang et al., 2016a). However, in contrast to these methods, our model has to use the language correction to improve, essentially implementing an interactive, user-guided reinforcement learning algorithm. As we will demonstrate in our experiments, iterative corrections cause this model to progressively improve its performance on the task very quickly. In the next section, we will describe a meta-learning algorithm that can train this model such that it is able to adapt to iterative corrections effectively at meta-test time.

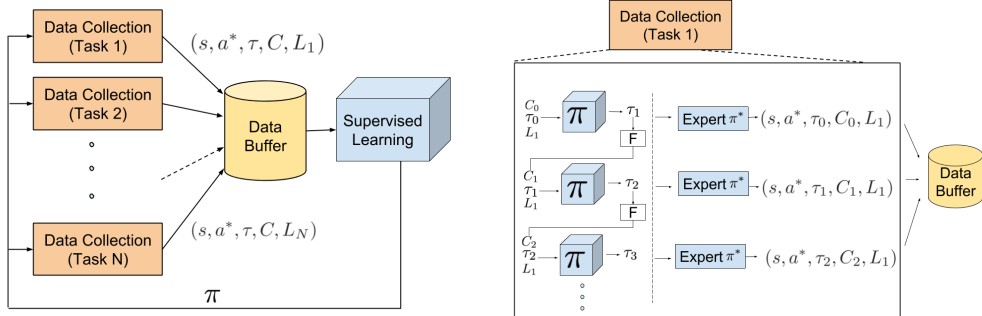

Figure 3: **Left:** Overall training procedure. We collect data for each task $[1, 2, \ldots, N]$ using DAgger, storing it in the data buffer. This is used to train a GPL policy with supervised learning. The trained policy is then used to collect data for individual tasks again, repeating the process until convergence. **Right:** Individual data collection procedure for a single task. The GPL policy is initially executed to obtain a trajectory $\tau_1$. This trajectory is corrected by an expert $\pi^*$ to generate data to be added to the buffer. The trajectory is then used to generate a correction, which is fed back into $\pi$, along with $\tau_1$ to generate a new trajectory $\tau_2$. This repeats until a maximum number of corrections is given, adding data to the buffer at each step.

## 5 META-TRAINING THE GPL MODEL TO LEARN FROM CORRECTIONS

In order for the GPL model to be able to learn behaviors from corrections, it must be meta-trained to understand both instructions and corrections properly, put them in the context of its own previous trajectories, and associate them with objects and events in the world. For clarity of terminology, we will use the term "meta-training" to denote the process of training the model, and "meta-testing" to denote its use for solving a new task with language corrections.

During meta-training, we assume access to samples from a distribution of meta-training tasks $T \sim p(T)$. The tasks that the model will be asked to learn at meta-test time are distinct from the meta-training tasks, though we assume them to be drawn from the same distribution, which is analogous to the standard distribution assumption in supervised learning. The tasks have the same state space $\mathcal{S}$ and action space $\mathbf{A}$, but each task has a distinct goal, and each task $T$ can be described by a different language instruction $L_T$. In general, more than one instruction could describe a single task, and the instructions might contain ambiguity.

Each of the tasks during meta-training has a ground truth objective provided by a reward function. We can use the reward function with any existing reinforcement learning algorithm to train a near-optimal policy. Therefore, we derive the algorithm for the case where we have access to a near-optimal policy $\pi_T^*(a|s)$ for each task $T$. In our experiments, $\pi_T^*(a|s)$ is obtained via reinforcement learning from ground truth rewards. For each meta-training task, we also assume that we can sample from the corresponding correction function $\mathcal{F}_T(c|\tau)$, which generates a correction $c$ for the trajectory $\tau$ in the context of task $T$. In practice, these corrections might be provided by a human annotator, though we use a computational proxy in our experiments. The key point is that we can use rewards to transform a collection of task-specific policies learned offline into a fast online learner during test time.

By using $\pi_T^*(a|s)$, $L_T$, and $\mathcal{F}_T(\tau)$, we can train our model for each task by using a variant of the DAgger algorithm (Ross et al., 2011), which was originally proposed for single-task imitation learning, where a learner policy is trained to mimic a near-optimal expert. We extend this approach to the setting of meta-learning, where we use it to meta-train the GPL model. Starting from an initialization where the previous trajectory $\tau_0$ and correction $c_0$ are set to be empty sequences, we repeat the following process: first, we run the policy corresponding to the current learned model $\pi(a|s, L_T, \boldsymbol{\tau}_0, \mathbf{c}_0)$ to generate a new trajectory $\tau_1$ for the task $T$. Every state along $\tau_1$ is then labeled with near-optimal actions by using $\pi_T^*(a|s)$ to produce a set of training tuples $(L_T, \tau_0, c_0, s, a^*)$. These tuples are appended to the training set $\mathcal{D}$. Then, a correction $c_1$ is sampled from $\mathcal{F}_T(c|\tau_1)$, and a new trajectory is sampled from $\pi(a|s, L_T, \boldsymbol{\tau}_1, \mathbf{c}_1)$. This trajectory is again labeled by the expert and appended to the dataset. In the same way, we iteratively populate the training set $\mathcal{D}$ with the states, corrections, and prior trajectories observed by the model, all labeled with near-optimal actions.

This process is repeated for a fixed number of corrections or until the task is solved, for each of the meta-training tasks. The model is then trained to maximum the likelihood of the samples in the dataset $\mathcal{D}$. Then, following the DAgger algorithm, the updated policy is used to again collect data for each of the tasks, which is appended to the dataset and used to train the policy again, until the process converges or a fixed number of iterations. This algorithm is summarized in Algorithm 1 and Figure 3.

## 6 LEARNING NEW TASKS WITH THE GPL MODEL

Using a GPL model meta-trained as described in the previous section, we can solve new "meta-testing" tasks $T_{test} \sim p(T)$ drawn from the same distribution of tasks with interactive language corrections. An initial instruction $L_T$ is first provided by the user, and the procedure for adapting with corrections follows the illustration in Figure 1. The learned policy is initially rolled out in the environment conditioned on $L_T$, and with the previous trajectory $\tau_0$ and correction $c_0$ initialized to the empty sequence. Once this policy generates a trajectory $\tau_1$, we can use the correction function $\mathcal{F}_T$ to generate a correction $c_1 = \mathcal{F}_T(\tau_1)$. The trajectory $\tau_1$, along with the correction $c_1$ gives us a new improved policy which is conditioned on $L_T$, $\boldsymbol{\tau}_1$, and $\mathbf{c}_1$. This policy can be executed in the environment to generate a new trajectory $\tau_2$, and the process repeats until convergence, thereby learning the new task. We provide the policy with the previous corrections as well but omit in the notation for clarity. This procedure is summarized in Algorithm 2.

This procedure is similar to meta-reinforcement learning (Finn et al., 2017; Duan et al., 2016), but uses grounded natural language corrections in order to guide learning of new tasks with feedback provided in the loop. The advantage of such a procedure is that we can iteratively refine behaviors quickly for tasks that are hard to describe with high level descriptions. Additionally, providing language feedback iteratively in the loop may reduce the overall amount of supervision needed to learn new tasks. Using easily available natural language corrections in the loop can change behaviors much more quickly than scalar reward functions.

---

**Algorithm 1:** GPL meta-training algorithm.

1   Initialize data buffer $\mathcal{D}$
2   **for** *iteration $j$* **do**
3     **for** *task $T$* **do**
4       Initialize $\tau_0 = 0$ and $c_0 = 0$
5       **for** *corr iter $i \in \{0, ..., c_{max}\}$* **do**
6         Execute $\pi(a|s, L_T, \boldsymbol{\tau}_i, \mathbf{c}_i)$ on $T$ to collect $\tau_{i+1}$
7         Obtain $c_{i+1} \sim \mathcal{F}_T(\tau_{i+1})$
8         Label $a^\star \sim \pi_T^\star(a|s), \forall\ s \in \tau_{i+1}$
9         Add all $(L_T, \tau_i, c_i, s, a^\star)$ to $D$
10     Train $\pi$ on $\mathcal{D}$.

---

**Algorithm 2:** Meta-testing: learning new tasks with the GPL model.

1   Given new task $T_i$, with instruction $L_T$
2   Initialize $\tau_0 = 0$ and $c_0 = 0$
3   **for** *corr iter $i \in \{0, ..., c_{max}\}$* **do**
4     Execute $\pi(a|s, L_T, \boldsymbol{\tau}_i, \mathbf{c}_i)$ on $T$ to collect $\tau_{i+1}$
5     Obtain $c_{i+1} \sim \mathcal{F}_T(\tau_{i+1})$

---

## 7 EXPERIMENTS

Our experiments analyze GPL in a partially observed object manipulation environment and a block pushing environment. The first goal of our evaluation is to understand where GPL can benefit from iterative corrections – that is, does the policy's ability to succeed at the task improve as each new correction provided. We then evaluate our method comparatively, in order to understand whether iterative corrections provide an improvement over standard instruction-following methods, and also compare GPL to an oracle model that receives a much more detailed instruction, but without the iterative structure of interactive corrections. We perform further experiments including ablations to evaluate the importance of each module and additional experiments with more varied corrections. We also compare our method to state of the art instruction following methods, other baselines which use intermediate rewards instead of corrections, and pretraining with language. Our code and supplementary material will be available at `https://sites.google.com/view/lgpl/home`

## 7.1 Experimental Setup

We describe the two experimental domains that we evaluate this method on. The first task is underspecified while the second task is ambiguous so each task lends itself to the use of corrections.

### 7.1.1 Multi-room Object Manipulation

Our first environment is discrete and represents the floor-plan of a building with six rooms (Figure 4), based on Chevalier-Boisvert & Willems (2018). The task is to pickup a particular object from one room and bring it to a goal location in another room. Each room has a uniquely colored door that must be opened to enter the room, and rooms contain objects with different colors and shapes. Actions are discrete and allow for cardinal movement and picking up and dropping objects. The environment is partially observed: the policy only observes the contents of an ego-centric 7x7 region centered on the present location of the agent, and does not see through walls or closed doors. The contents of a room can only be observed by first opening its door.

This environment allows for the natural language instruction which specifies the task to be underspecified. The instruction is given as "Move <goal object color> <goal object shape> to <goal square color> square". Since the agent cannot see into closed rooms, it does not initially know the locations of the goal object or goal square. It must either explore or rely on external corrections which guide the agent to the appropriate rooms.

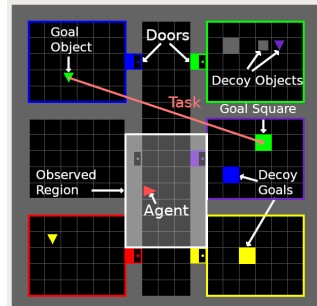

Environments are generated by sampling a goal object color, goal object shape, and goal square color which are placed at random locations in different random rooms. There are 6 possible colors and 3 possible object shapes. Decoy objects and goals are placed randomly among the six rooms. The only shared aspect between tasks are the color of the doors so the agent must learn to generalize across a variety of different objects across different locations.

Figure 4: The multi-room object manipulation environment with labeled components.

To generate the corrections, we describe a task as a list of subgoals that the agent must complete. For example, the instruction in Figure 4 is "Move green triangle to green square", and the subgoals are "enter the blue room", "pick up the green triangle", "exit the blue room", "enter the purple room", and "go to the green goal". The correction for a given trajectory is then the first subgoal that the agent failed to complete. The multistep nature of this task also makes it challenging, as the agent must remember to solve previously completed subgoals while incorporating the corrections to solve the next subgoal.

The training and test tasks are generated such that for any test task, its list of all five subgoals does not exist in the training set. There are 3240 possible lists of all five subgoals. We train on 1700 of these environments and reserve a separate set for testing.

### 7.1.2 Robotic Object Relocation

The second environment is a robotic object relocation task built in Mujoco (Todorov et al., 2012) shown in Figure 5. The task is to apply force on a gripper to push one of three blocks to a target location. Five immovable blocks scattered in the environment act as obstacles. States are continuous and include the agent and block positions. Actions are discrete and apply directional forces on the gripper.

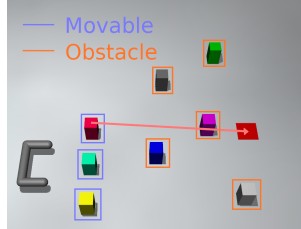

The instruction is given as "Move <goal block color> close to <obstacle block color>," where the closest or $2^{nd}$ closest obstacle block to the target location is randomly chosen. This instruction is ambiguous, as it does not describe the precise location of the target. Corrections will help guide the agent to the right location relative to the obstacles and previous actions of the agent. Environments are generated by sampling one of the three movable blocks to push and sampling a random target location. We generate 1000 of these environments and train on 750 of

Figure 5: The robotic object relocation environment. The agent must push the red block right of the magenta block.

them. There are three types of corrections, and feedback is provided
by stochastically choosing between these types. The correction types are directional ("Move a little left", "Move a lot down right"), relational ("Move left of the white block", "Move closer to the pink block"), or indicate the correct block to push ("Touch the red block").

This environment was chosen because of its continuous state space and configuration of objects which allows for spatial relations between objects. It is more natural and convenient for a user to specify a goal with language, e.g., "close to the pink block," than using exact spatial coordinates. Furthermore, because the corrections can be relative to other blocks, the policy must learn to contextualize the correction in relation to other objects and its own previous behavior.

## 7.2 COMPARISONS

We compare our method to two types of alternative methods - 1) alternative methods for instruction following 2) methods using rewards for finetuning on new tasks.

We first compare our method to a standard instruction following method using our architecture, a method that receives full information, and to an instruction following architecture from Misra et al. (2017b) which we call MIVOA. We test MIVOA with both the instruction and full information. All methods are trained with DAgger. The instruction following methods only receive the instruction, which is ambiguous and does not contain the precise locations of objects or target positions. The full information methods receive the exact information the agent needs but for which a human may not always want to provide or have access to. For the multi-room environment the full information method receives *all* the subgoals that are needed to solve the task, but does not receive them interactively. The full information method for the robotic object relocation task receives which block to move and the exact coordinates of the goal and so is an oracle. We measure the performance of an agent on the task by computing the completion rate: the fraction of subgoals (max 5) that the agent has successfully completed. The completion rate for the robotic domain is $1 - \frac{\text{final block dist}}{\text{initial block dist}}$. We expect our model to perform better than the instruction following baseline as it can receive the missing information through the corrections and we expect it to come close to performing as well as the full information method. The results for this comparison is shown in Table 1.

In the second set of comparisons, we compare the sample complexity of learning new tasks with GPL against other baselines which utilize the reward function for learning new tasks (Figure 6). We compare against a reinforcement learning (RL) baseline that trains a separate policy per task using the reward. In addition, we run a baseline that does pretraining with DAgger on the training tasks and then finetunes on the test tasks with RL, thereby following a similar procedure as Andreas et al. (2018). In both domains, the pretrained policy receives the instruction. For RL and finetuning we use the same algorithm (Schulman et al., 2017) and reward function used to train the expert policies. Additionally, in order to evaluate if the language corrections provide more information than just a scalar correction we also run a version of our method called GPR (Guiding Policies with Reward) which replaces the language correction with the reward. The correction for a trajectory is the sum of rewards of that trajectory. The performance of all of these methods is shown in Figure 6.

## 7.3 LEARNING NEW TASKS QUICKLY WITH LANGUAGE CORRECTIONS

As described above, we consider two domains - multi-room object manipulation and a robotic object relocation task. In the object relocation domain, the test tasks here consist of new configurations of objects and goals. For the robotic domain, the test tasks consist of pushing to new target locations. Details on the meta-training complexity of our method are in appendix A.3.

In the first set of comparisons mentioned in Section 7.2, we measure the completion rate of our method for various numbers of corrections on the test tasks. The instruction baseline does not have enough information and is unable to effectively solve the task. As expected, we see increasing completion rates as the number of corrections increases and the agent incrementally gets further in the task. For the multi-room domain our method matches the full information baseline with 3 corrections and outperforms it with 4 or more corrections. Since the full information baseline receives all 5 subgoals, this means our method performs better with *less* information. The interactive nature of our method allows it to receive only the information it needs to solve the task. Furthermore, our model must learn to map corrections to changes in behavior which may be more modular, disentangled, and

easier to generalize compared to mapping a long list of instructions to a single policy that can solve the task. For the robotic domain, our model exceeds the performance of the instruction baseline with just 1 correction. With more corrections it comes close to the full information method which receives the exact goal coordinates.

In the second set of comparisons mentioned in Section 7.2, we compare against a number of baselines that use the task reward (Fig 6). We see that our method can achieve high completion rate with very few test trajectories. While GPL only receives up to five trajectories on the test task, the RL baseline takes more than 1000 trajectories to reach similar levels of performance. The RL baseline is able to achieve better final performance but takes orders of magnitude more training examples and has access to the test task reward. The pretraining baseline has better sample complexity than the RL baseline but still needs more than 1000 test trajectories. The reward guided version of our method, GPR, performs poorly on the multi-room domain but obtains reasonable performance for the robotic domain. This may indicate that language corrections in the multi-room domain provide much more information than just scalar rewards. However, using scalar rewards or binary preferences may be an alternative to language corrections for continuous state space domains such as ours.

| Env | Instruction | Full Info | MIVOA (Instr.) | MIVOA (Full Info) | $c_0$ | $c_1$ | $c_2$ | $c_3$ | $c_4$ | $c_5$ |
|---|---|---|---|---|---|---|---|---|---|---|
| Multi-room | 0.075 | 0.73 | 0.067 | 0.63 | 0.066 | 0.46 | 0.65 | 0.73 | 0.77 | **0.82** |
| Obj Relocation | 0.64 | **0.96** | 0.65 | - | 0.65 | 0.80 | 0.84 | 0.85 | 0.88 | 0.90 |

Table 1: Mean completion rates on test tasks for baseline methods and ours across 5 random seeds. $c_i$ denotes that the agent has received $i$ corrections. GPL is able to quickly incorporate corrections to improve agent behavior over instruction following with fewer corrections than full information on the multi-room domain. MIVOA is an architecture from Misra et al. (2017b).

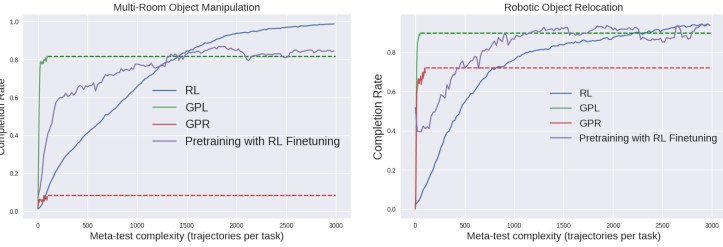

Figure 6: Sample complexity on test tasks. The mean completion rate is plotted against the number of trajectories using during training per task. Our method (GPL) is shown in green.

### 7.4 ANALYZING THE BEHAVIOR OF GPL

| Ablations | $c_0$ | $c_1$ | $c_2$ | $c_3$ | $c_4$ | $c_5$ |
|---|---|---|---|---|---|---|
| Base | 0.066 | 0.46 | 0.65 | 0.73 | 0.77 | 0.82 |
| No instruction | 0.059 | 0.45 | 0.62 | 0.72 | 0.78 | 0.79 |
| No trajectory | 0.077 | 0.44 | 0.62 | 0.70 | 0.76 | 0.77 |
| Only immediate correction | 0.067 | 0.49 | 0.44 | 0.58 | 0.59 | 0.63 |

Table 2: Ablation Experiments analyzing the importance of various components of the model on the multi-room env. We see that removing previous corrections (only $c_i$) performs the worst, while removing instruction $L$ is less impactful.

To understand GPL better, we perform a number of different analyses - model ablations, extrapolation to more corrections, more varied corrections and generalization to out of distribution tasks.

We perform ablations on the multi-room domain to analyze the importance of each component of our model in Figure 2. For the three ablations, we remove the instruction $L$, remove the previous trajectory $\tau_i$, and provide only the immediate previous correction $c_i$ instead of all previous corrections. We find that removing the instruction hurts the performance the least. This makes sense because the model can receive the information contained in the instruction through the corrections. Removing

the previous corrections hurts the performance the most. Qualitatively, the agent that does not have access to the previous tends to forget what it had done previously and erases the progress it made. This explains the dip in performance from $c_1$ to $c_2$ for the only immediate correction.

We also investigate if performance continues to increase if we provide more corrections at meta-testing time than seen during training. In Table 3, we provide up to 10 corrections during test time while only up to 5 corrections are seen during training. We see that completion rate increases from $0.82$ to $0.86$ for the multi-room domain and from $0.90$ to $0.95$ for the object relocation domain. These are small gains in performance and we expect to see diminishing returns as the number of corrections increases even further.

| Env | $c_5$ | $c_7$ | $c_{10}$ |
|---|---|---|---|
| Multi-room | 0.82 | 0.83 | 0.86 |
| Obj Relocation | 0.90 | 0.91 | 0.95 |

Table 3: Mean completion rates on test tasks for 5, 7, 10 corrections. Only up to 5 corrections are seen during training.

In Table 4 we investigate what happens if we give an agent more varied corrections in the multi-room environment. We add two new correction types. The first is directional and specifies which of the eight cardinal direction the next goal is in, e.g. "goal room is southwest." The second type is binary and consists of simple "yes/no" type information such as "you are in the wrong room".

| Type | $c_1$ | $c_2$ | $c_3$ | $c_4$ | $c_5$ |
|---|---|---|---|---|---|
| Directional | 0.242 | 0.343 | 0.43 | 0.51 | 0.56 |
| Binary | 0.073 | 0.078 | 0.08 | 0.089 | 0.09 |
| All | 0.236 | 0.363 | 0.44 | 0.538 | 0.606 |

Table 4: Experiments investigating different correction types and effect on performance (mean completion). The experiments agree with intuition that binary carries little information and results in a small increase of the completion rate. Directional corrections which gives an intermediate amount of information result in a fair increase in performance, but less than fully specified correction.

We also experiment with if our method can generalize to unseen objects in the multi-room domain. We holdout specific objects during training and test on these unseen objects. For example, the agent will not see green triangles but will see other green objects and non-green triangles during training and must generalize to the unseen combination at test time. In Table 5, we see that our method achieves a lower completion rate compared to when specific objects are not held out but is still able to achieve a high completion rate and outperforms the baselines.

| Env | Full Info | MIVOA (Full Info) | $c_0$ | $c_1$ | $c_2$ | $c_3$ | $c_4$ | $c_5$ |
|---|---|---|---|---|---|---|---|---|
| Multi-room | 0.57 | 0.62 | 0.073 | 0.44 | 0.58 | 0.65 | 0.73 | 0.75 |

Table 5: Experiments on holding out specific objects during training to see if our method can generalize to unseen objects at test time.

## 7.5 Discussion and Future Work

We presented meta-learning for guided policies with language (GPL), a framework for interactive learning of tasks with in-the-loop language corrections. In GPL , the policy attempts successive trials in the environment, and receives language corrections that suggest how to improve the next trial over the previous one. The GPL model is trained via meta-learning, using a dataset of other tasks to learn how to ground language corrections in terms of behaviors and objects. While our method currently uses fake language, future work could incorporate real language at training time. To scale corrections to real-world tasks, it is vital to handle new concepts, in terms of actions or objects, not seen at training time. Approachs to handling these new concepts could be innovations at the model level, such as using meta-learning, or at the interface level, allowing humans to describe new objects to help the agent.

## ACKNOWLEDGEMENTS

This work was supported by the Office of Naval Research through a Young Investigator Program Award, Berkeley Deep Drive, the National Science Foundation through IIS-1614653, and computational resources from Amazon and NVIDIA.

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

# A APPENDIX

## A.1 VISUALIZING BEHAVIOR: MULTI-ROOM OBJECT MANIPULATION ENVIRONMENT

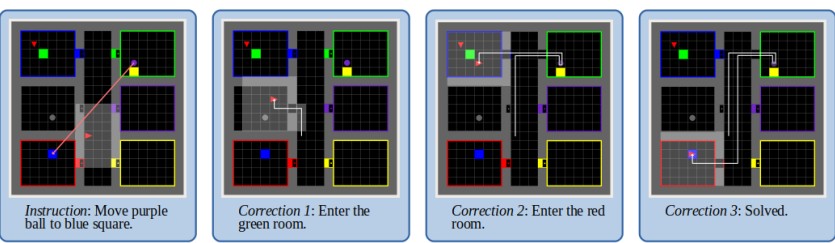

Figure 7: Example task with corrections. Instruction: The agent receives the initial instruction. Correction 1: The agent mistakenly goes into the gray door, so it receives the correction to enter the green room, where the purple ball is located. Correction 2: The agent successfully picks up the ball, but then mistakenly enters the blue room, so it receives the correction to enter the red room, where the goal is located. Correction 3: The agent brings the object to the goal and solves the task.

An example task with corrections in show in Figure 7.

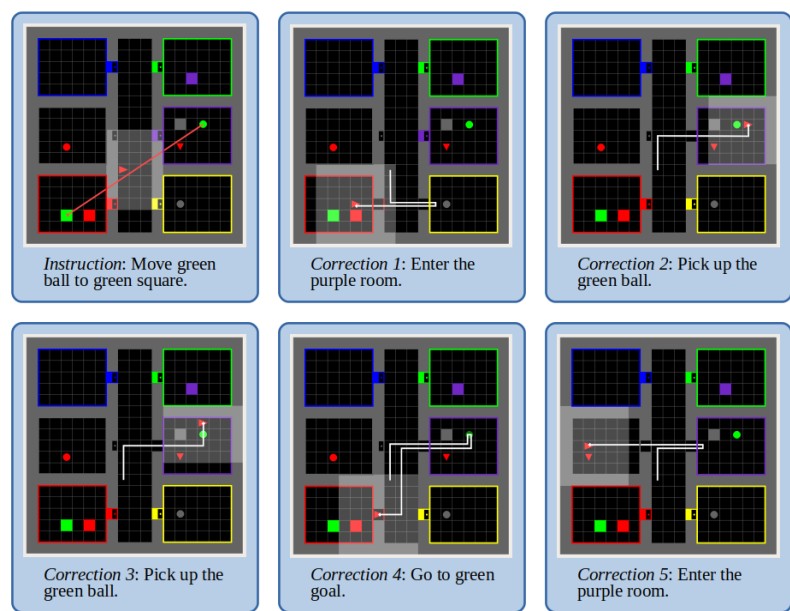

Figure 8: Failure example. The orange arrow shows the task, the white arrows show the net trajectory.

It is possible to visualize failure cases, which illuminate the behavior of the algorithm on challenging tasks. In the failure case in Figure 8, we note that the agent is able to successfully enter the purple room, pickup the green ball, and exit. However, after it receives the fourth correction telling it to go to the green goal, it forgets to pick up the green ball. This behavior can likely be improved by varying corrections more at training time, and providing different corrections if an agent is unable to comprehend the first one.

Additionally, we present a success case in Figure 9, where the agent successfully learns to solve the task through iterative corrections, making further progress in each frame.

## A.2 VISUALIZING BEHAVIOR: ROBOTIC OBJECT RELOCATION ENVIRONMENT

Similarly, we can again visualize failure cases for the robotic object relocation environment. In the failure case in Figure 10, the agent pushes the correct object, and very nearly solves the task after the

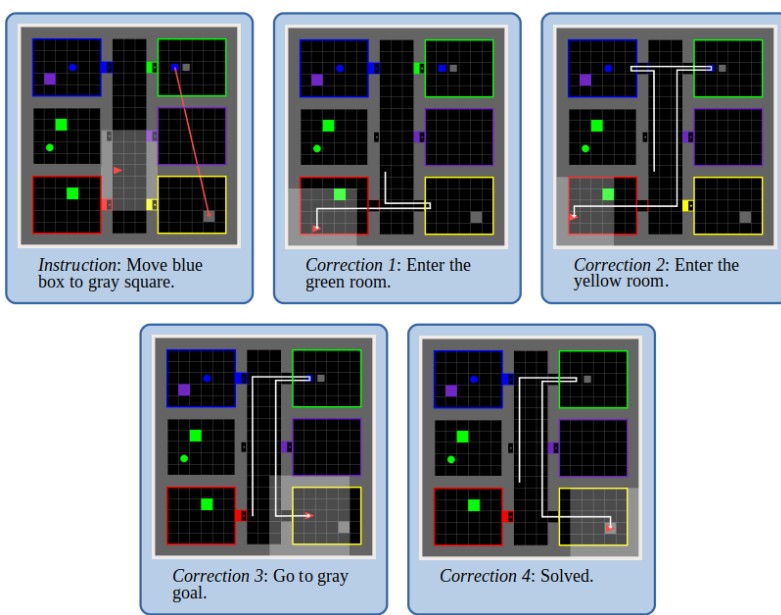

Figure 9: Success example. The orange arrow shows the task.

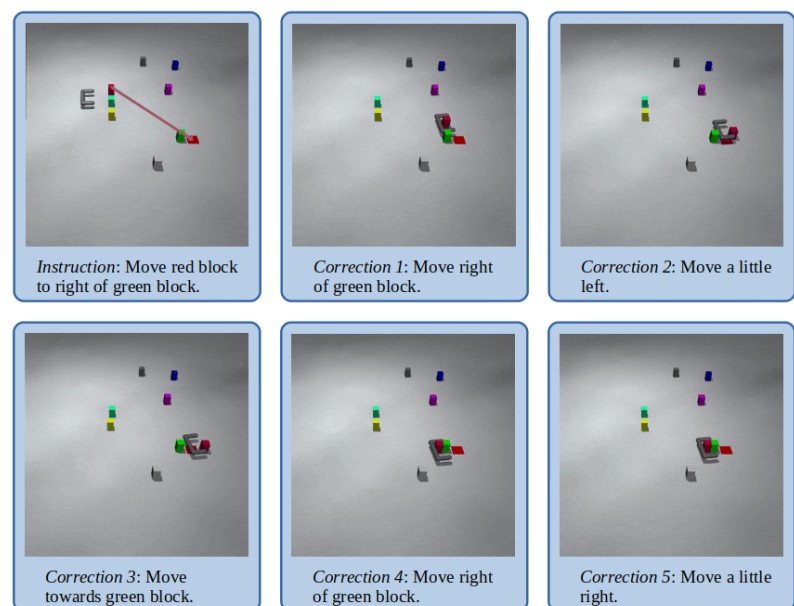

Figure 10: Failure example. The orange arrow shows the task, the white arrows show the net trajectory.

first and second corrections. However, after it receives the third and fourth corrections telling it to go near the green block, its trajectory changes such that its path to the goal is blocked by the green block itself. This case is particularly challenging, as the combination of the bulky agent body, angle of approach, and the proximity of the goal location to the obstacle near it make the goal location inaccessible. Were the goal location a little further from the green block, or the agent of a form lending itself to nimbler motion, this task may have been solved by the second correction.

We also present a success case in Figure 11, where the agent starts off pushing the wrong block, but successfully learns to solve the task through iterative corrections, making further progress in each frame.

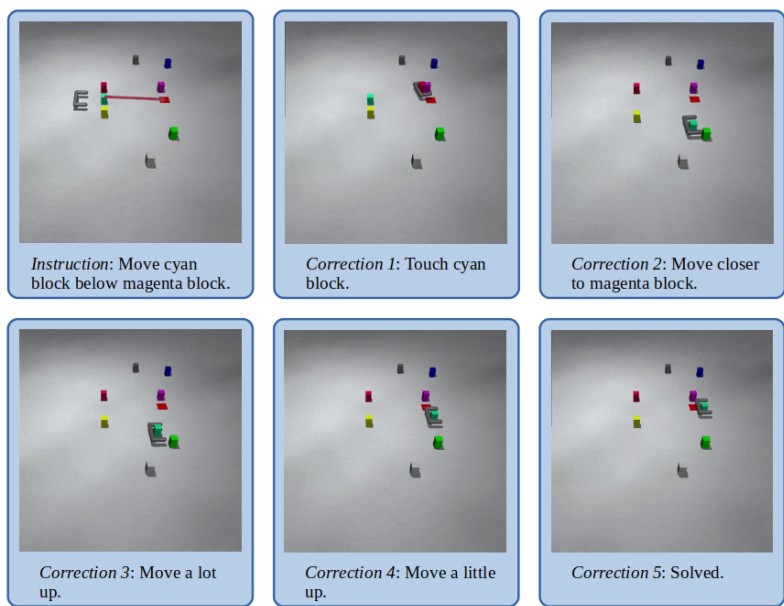

Figure 11: Success example. The orange arrow shows the task.

## A.3 TRAINING AND ARCHITECTURE DETAILS

We detail the training and architecture details for each environment. We use Adam for optimization with a learning rate of 0.001. The data buffer size is $1e6$. We train on the whole data buffer for five epochs each time before adding more data to the buffer. Unless otherwise stated, we use ReLU activations. MLP(32, 32) specifies a multilayer-perceptron with 2 layers each of size 32. CNN((4, 2x2, 1), (4, 2x2, 1)) specifies a 2 layer convolutional neural network where each layer has 4 filters, 2x2 kernels, and 1 stride.

To train the expert policies we use Proximal Policy Optimization with a dense reward function. For the multi-room domain we keep track of the next subgoal the agent has to complete. The reward is $-0.01*$ Euclidian distance to the next subgoal where a grid is a distance of 1. If the agent completes a previously uncompleted subgoal it gets a onetime reward of 100. For the robotic object manipulation domain the reward is -(Euclidian distance of the gripper to the block $+ 5*$ Euclidian distance of the block to the target location).

For the multi-room domain we meta-train on 1700 environments. Our method converges in 6 DAgger steps so it takes 30 corrections per environment for a total of 51,000 corrections. For the robotic object relocation domain, we train on 750 environments. Our method converges in 9 DAgger steps so it takes 45 corrections per environment for a total of 33,750 corrections.

### A.3.1 MULTI-ROOM OBJECT MANIPULATION

The observation is a 7x7x4 grid centered on the agent. The $1^{st}$ channel encode object types (empty, wall, closed door, locked door, triangle, square, circle, goal). The $2^{nd}$ channel encodes colors (none, blue, green, gray, purple, red, yellow). The $3^{rd}$ and $4^{th}$ channels also encode object types and colors respectively but only for objects the agent is currently holding. The observation also includes a binary indicatory if the agent is currently holding an object. The action space is of size 6 and consists of (move up, move left, move right, move down, pickup object, drop object). The length of a trajectory is 100.

**Correction Module** Each previous trajectory is first subsampled every $25^{th}$ state. Each observation is then processed by a 2D CNN((4, 2x2, 1), (4, 2x2, 1)) followed by a MLP(32, 32). This trajectory of state embeddings is then processed by a 1D CNN(4, 2, 1) followed by a MLP(16, 4).

The language correction is converted into word embeddings of size 16 and processed by a 1D CNN(4, 2, 1) followed by a MLP (16, 4). For a given correction iteration, the trajectory embedding and the

correction embedding are then concatenated and fed through a MLP(32 ,32). These tensors are then averaged across the number of correction iterations.

**Instruction Module** The instruction is converted into word embeddings (the same embeddings as the correction) of size 16 and processed by a 1D CNN(4, 2, 1) followed by a MLP(16, 32).

**Policy Module** The observation is processed by a 2D CNN((8, 2x2, 1), (8, 2x2, 1)) followed by a MLP(32, 32). This state embedding, the correction module tensor, and the instruction module tensor are concatenated and fed through a MLP (64, 64) to output an action distribution.

### A.3.2 ROBOTIC OBJECT RELOCATION

The observation is of size 19 and includes the (x, y, orientation) of the gripper and the (x, y) coordinates of the 3 movable blocks and 5 fixed blocks. The action space is of size 4 and consists of applying force on the gripper in the cardinal directions. The length of a trajectory is 350.

**Correction Module** Each previous trajectory is first subsampled every $70^{th}$ state. The trajectory is then processed by a 1D CNN(8, 2, 1) followed by a MLP(16, 8).

The language correction is converted into word embeddings of size 16 and processed by a 1D CNN(4, 2, 1) followed by a MLP(16, 16). For a given correction iteration, the trajectory embedding and the correction embedding are then concatenated and fed through a MLP(32 ,32). These tensors are then averaged across the number of correction iterations.

**Instruction Module** The instruction is converted into word embeddings (the same embeddings as the correction) of size 16 and processed by a 1D CNN(4, 2, 1) followed by a MLP(16, 16).

**Policy Module** The observation, the correction module tensor, and the instruction module tensor are concatenated and fed through a MLP(256, 256, 256) to output an action distribution.

