# OpenReview forum: "Guiding Policies with Language via Meta-Learning"
_ICLR.cc/2019/Conference_

### Official Review · AnonReviewer1 · 2018-10-25
**Nice idea, very limited experimental validation**

**Rating:** 6
**Confidence:** 4

**Review:**



UPDATE: I've increased my rating based on the authors' thorough responses and the updates they've made to the paper. However, I still have a concern over the static nature of the experimental environments.

=====================

This paper proposes the use of iterative, linguistic corrections to guide (ie, condition and adjust) an RL policy. A major challenge in learning language-guided policies is grounding the language in environment states and agent actions. The authors tackle this challenge with a meta-learning approach.

The approach is fairly complex, blending imitation and supervised learning. It operates on a training set from a distribution of virtual pick-move-place tasks. The policy to be learned operates on this set and collects data, via something close to DAgger, for later supervised learning on the task distribution. The supervised-learning data comprises trajectories augmented with linguistic subgoal annotations, which are referred to as policy "corrections." By ingesting its past trajectories and the correction information, the policy is meant to learn to solve the task and to ground the corrections at the same time, end-to-end. Correction annotations are derived from an expert policy.

The idea of guiding a policy through natural language and the requisite grounding of language in environment states and policy actions have been investigated previously: for example, by supervised pretraining on a language corpus, as in the cited work of Andreas et al. (2018). The alternative meta-learning approach proposed here is both well-motivated and original.

Generally, I found the paper clear and easy to read. The authors explain convincingly the utility of guiding policies through language, especially with respect to the standard mechanisms of reward functions (sparse, engineered) and demonstrations (expertise required). The paper is also persuasive on the utility of iterative, interactive correction versus a fully-specified language instruction given a priori. The meta-learning algorithm and training/test setup are both explained well, despite their complexity. On the other hand, most architectural details necessary to reproduce the work are missing, at least from the main text. This includes various tensor dimensions, the structure of the network for perceiving the state, etc.

I like the proposed experimental setting. It enables meta-learning on sequential decision making problems in a partially observable environment, which seems useful to the research community at large. Ultimately, however, this paper's significance is not evident to me, mainly because the proposed method lacks thorough experimental validation. No standard baselines are evaluated on the task (with or without meta-learning), nor is a detailed analysis of the learned policies undertaken. The ablation study is useful, and a good start, but insufficient in my opinion. Unfortunately, the results are merely suggestive rather than convincing.

Some things I'd like to see in an expanded results section before recommending this paper include:
- Comparison to an RL baseline that attempts to learn the full task, without meta-training or language corrections.
- Comparison to a baseline that learns from intermediate rewards. Instead of annotating data with corrections, you could provide +/- scalar rewards throughout each trajectory based on progress towards the goal (since you know the optimal policy). How effective might this be compared to using the corrections?
- Comparison to a baseline that does some kind of pretraining on the language corrections, as in Andreas et al. (2018).
- Quantification of how much meta-training data is required. What is the sample complexity like with/without language corrections?

I also have concerns about the need for near-optimal agents on each task -- this seems very expensive and inefficient. The expert policy is obtained via RL on each individual task using "ground truth" rewards. It is not specified what these rewards are, nor is it stated how near to optimal the resulting policy is nor how this nearness affects the overall meta-learning process.

Its unclear to me how the "full information" baseline processes and conditions on the full set of subgoals/corrections. Are they read as a single concatenated string converted to one vector by the bi-LSTM?

There also might be an issue with the experimental setup, unless I've misunderstood it. The authors state that "the agent only needs 2 corrections where the first correction is the location of the goal object and the second is the location of the goal square." But if the specific rooms, indicated by colors, do not change location from task to task (and they appear not to from all the figures), then the agent can learn the room locations during meta-training and these two "corrections" tell it everything it needs to know to solve the task.

Pros:
- Appealing, well-motivated idea for training policies via language.
- Clear, pleasant writing and good communication of a complicated algorithm.
- Good experimental setup that should be useful in other research (except for possible issue with static room locations).

Cons:
- The need for a near-optimal policy for each task.
- Overall complexity of the training process.
- The so-called corrections are actually linguistic statements of subgoals computed from the optimal policy. There is much talk in the introduction of interactive policy correction by humans, which is an important goal and interesting problem, but the present paper does not actually investigate human interaction. This comes as a letdown after the loftiness of the introduction.
- Various details needed for reproduction are lacking. Maybe they're in the supplementary material; if so, please state that in the main text.
- Major lack of comparisons to alternative approaches.

---

> ### Author Response · Authors · 2018-11-16
> **R1 Review Response (Part 1)**
>
> Thank you for the detailed and constructive feedback. To address concerns about limited experimental evaluation, we have added a new environment that we call robotic object relocation, which involves a continuous state space and more relative corrections. The results for this environment are in the revised paper in Section 7.3. To address comments about comparisons, we have also added a number of additional comparisons, comparing LGPL to state of the art instruction following methods (Misra 2017, Table 1), pre-training with language (similar to Andreas 2018, Fig 7), using rewards instead of language corrections (Fig 7), and training from scratch via RL (Fig 7). Additionally, to provide a deeper understanding of the methods performance, we included a number of additional analyses on the methods extrapolation and generalization in Section 7.4.
>
> We would appreciate it if the reviewer could take another look at our changes and additional results, and let us know if they would like to either revise their rating of the paper, or request additional changes that would alleviate their concerns.
>
> Find below the responses to specific comments:
>
> “No standard baselines are evaluated on the task (with or without meta-learning), nor is a detailed analysis of the learned policies undertaken. “
> -> We have added additional comparisons and points of analysis to the updated paper. We compare with a strong instruction following method from the literature, Misra et al. (2017) (Table 1), as well as a number of other comparisons including all the comparisons that were requested (see detailed comments below) (Fig 7).
>
> We have also added a number of new points of analysis. We analyze the performance of the method on stochastically chosen corrections instead of very well formed ones (Table 4). We analyze the extrapolation performance of the method to more corrections than training time  (Table 3). We also analyze the performance of LGPL on tasks that are slightly out of distribution (Table 5). We would be happy to add additional analysis that the reviewer believes is important for the paper -- please let us know if we have addressed all of your concerns in this regard!
>
> “- Comparison to an RL baseline that attempts to learn the full task, without meta-training or language corrections.”
> > We have added a RL baseline that trains a separate policy per task using a dense reward (Section 7.3, Fig 7). The details of the reward functions and training algorithm can be found in the appendix A.3. The RL baseline is able to achieve better final performance but takes orders of magnitude more samples on the new tasks. Our method can obtain reasonable performance with just 5 samples on the test tasks. An important distinction to make is that this baseline also assumes access to the test task reward function; our method only uses the language corrections. Additional details can be found in Section 7.3, 7.4.
>
> “Comparison to a baseline that learns from intermediate rewards. Instead of annotating data with corrections, you could provide +/- scalar rewards”
> > We have added a baseline (Section 7.3, Fig 7) that uses intermediate rewards instead of language corrections, that we call Reward Guided Policy Learning (RGPL).The correction for a trajectory is the sum of rewards of that trajectory. RGPL performed worse than LGPL in both domains as seen in Fig 7. As seen from Fig 7 language corrections allow for more information to be transmitted over scalar rewards. Additional details for this comparison can be found in Section 7.2.
>
> “- Comparison to a baseline that does some kind of pretraining on the language corrections, as in Andreas et al. (2018).”
> > We have added a baseline (Section 7.3, Fig 7) that follows a pre-training paradigm similar to Andreas et al (2018) --  first pre-train a model on language instructions across many tasks and then finetune the model on new tasks using task-specific reward. Andreas et al. (2018) trains a learner with task-specific expert policies using DAgger. It then searches in the instruction space for the policy with the highest reward and then adapts the policy to individual tasks by fine tuning with RL. Since we can provide the exact instruction the policy needs, we do not perform the search in instruction space. We pretrain on the training tasks with DAgger and then finetune on test tasks with RL. This baseline is able to achieve slightly better final performance both domains but takes orders of magnitude more samples on the test tasks (>1000 trajectories vs 5 for our method). Details for this comparison can be found in Section 7.3.

---

> > ### Author Response · Authors · 2018-11-16
> > **R1 Review Response (Part 2)**
> >
> > “- Quantification of how much meta-training data is required. What is the sample complexity like with/without language corrections?”
> > > We add these details to the paper in Appendix A.3.
> >
> > Meta-training: For the multi-room domain we meta-train on 1700 environments. Our method converges in 6 DAgger steps so it takes 30 corrections per environment for a total of 51,000 corrections. For the robotic object relocation domain, we train on 750 environments. Our method converges in 9 DAgger steps so it takes 45 corrections per environment for a total of 33750 corrections.
> >
> > Meta-testing: On new tasks, asymptotically RL is able to achieve better final performance than our method but takes orders of magnitudes more samples. In Figure 7 we plot the number of training trajectories used per test task. While LGPL only receives up to 5 trajectories for each test task, RL takes more than 1000 trajectories to reach similar levels of performance.
> >
> > “Its unclear to me how the "full information" baseline processes and conditions on the full set of subgoals/corrections. Are they read as a single concatenated string converted to one vector by the bi-LSTM?”
> > > For the full information baseline, all the subgoals are concatenated and converted to one vector by a bi-LSTM.
> >
> > “I also have concerns about the need for near-optimal agents on each task -- this seems very expensive and inefficient.”
> > > To ground the language corrections we need some form of supervision. Typically methods for grounding natural language instructions assume access to either a large corpus of supervised data (i.e expert behavior) or a reward function (Janner et al 2017, Misra et al 2017, Wang, Xiong, et al. 2018, Andreas et al) in order to train the model. In our setting, we similarly assume access to near optimal agents or a reward function (which we can use to train near optimal agents), which is used to learn the policy and language grounding, but only on the meta-training tasks. On unseen meta-test tasks, we can learn very quickly simply by using language corrections, without the need for reward functions or expert policies.
> >
> > “On the other hand, most architectural details necessary to reproduce the work are missing, at least from the main text.”
> > > We have added architecture and training details (including reward functions) to the appendix A.3 and referenced them in the main text. We also intend to open source the code once the review decision is out.
> >
> >
> > [1] Wang, Xin et al. “Look Before You Leap: Bridging Model-Free and Model-Based Reinforcement Learning for Planned-Ahead Vision-and-Language Navigation.” CoRRabs/1803.07729 (2018)
> >
> > [2] Andreas, Jacob et al. “Learning with Latent Language.” NAACL-HLT (2018).
> >
> > [3] Misra, Dipendra Kumar et al. “Mapping Instructions and Visual Observations to Actions with Reinforcement Learning.” EMNLP (2017).
> >
> > [4] Janner, Michael et al “Representation Learning for Grounded Spatial Reasoning” TACL 2017

---

> > > ### Comment · AnonReviewer1 · 2018-11-19
> > > **Thanks, but one more question!**
> > >
> > > I'm very impressed and mostly satisfied with the responses to my review. There remains one important, unanswered question, however, that I'd like to be addressed.
> > >
> > > If the specific rooms, indicated by colors, do not change location from task to task (and they appear not to from all the figures), then the agent can learn the room locations during meta-training and the two "corrections" tell it everything it needs to know to solve the task. So: do colored rooms change location from task to task? I.e., is the blue room sometimes in the lower right and other times in the upper left, etc?

---

> > > > ### Author Response · Authors · 2018-11-22
> > > > **Experiment Clarification**
> > > >
> > > > For the multi-room environment, the room colors do not change location from task to task. While two corrections could tell it everything it needs to know and we observe this in some cases, we see that the agent often fails to complete subgoals it has information on and still benefits from successive corrections after two (as seen in Table 1). An example of this can be seen in Appendix A.1. Here the agent is not perfect and is able to complete the task after receiving multiple corrections (sometimes the same correction twice).
> > > >
> > > > Our model is also able to handle more relative types of corrections where the agent cannot memorize absolute positions. In Section 7.4 we add different types of corrections such as (“you are in the wrong room” or “goal room is southwest.” The agent cannot just memorize the locations of each room and instead must map corrections to changes in behavior.
> > > >
> > > > The second environment we have added, the robotic object relocation task, has relative corrections such as “Move a little up right” or “Push closer to the green block”. A fixed number of corrections cannot exactly specify the task and the agent must consider the correction in terms of its previous behavior to gradually move closer to the goal.

---

> > > > > ### Comment · AnonReviewer1 · 2018-11-22
> > > > > **Thanks for the clarification**
> > > > >
> > > > > Based on your thorough responses and paper modifications, I'll revise my review.

---

### Official Review · AnonReviewer3 · 2018-11-05
**Meta-Learning Language-Guided Policy Learning**

**Rating:** 6
**Confidence:** 3

**Review:**

Summary:
This paper studies how to teach agents to complete tasks via natural language instructions in an iterative way, e.g., correct the behavior of agents. This is a very natural way to learn as humans. The basic idea is to learn a model that takes correction and history as inputs and output what action to take. This paper formulates this in meta-learning setting in which each task is drawn from a pre-designed task distribution and then the models are able to adapt to new tasks very fast. The proposed method is evaluated in a virtual environment where the task is to pick up a particular object in a room and bring it to a particular goal location in a different room. There are two baselines: 1) instruction only (missing information), 2) full information (not iterative), the proposed method outperforms 1) with higher task completion rate and 2) with fewer number of corrections.

Strength:
- This paper addresses a very interesting problem in order to make agents learn more human like.

Comments:
- Only one setting is studied. And, the task distribution seems not very complex.
- How the proposed model performs if the task is a little bit out of distribution?

---

> ### Author Response · Authors · 2018-11-16
> **R3 Review Response**
>
> Thank you for the detailed and constructive feedback. We have made a number of changes to the paper to address this feedback - including new experimental domains, more comparisons and in-depth analysis of model behavior. We describe these further in responses to specific comments below:
>
> “Only one setting is studied”
> > To extend the experimental evaluation beyond a single domain, we have added a new environment that we call robotic object relocation and involves manipulating a robotic gripper to push blocks. This environment involves relative corrections and continuous state space and is described in Section 7.1.2. This environment shows our method can generalize to substantially different domains (continuous state space) as well as new kinds of corrections beyond subgoals. The results for this environment are in the revised paper in Section 7.3.
>
> “the task distribution seems not very complex.”
> > We specify the task distribution in Section 7.1. For the multi-room environment the training and test tasks are generated such that for any test task, its list of all five subgoals does not exist in the training set. There are 3240 possible lists of all five subgoals. We train on 1700 of these environments and reserve a separate set for testing. For the robotic object relocation environment, we generate tasks by sampling one of the 3 movable blocks to be pushed. We then randomly choose one of the 5 immovable blocks and sample a direction and distance from that block to get a goal location. We generate 1000 of these environments and train on 750 of them.
>
> “How the proposed model performs if the task is a little bit out of distribution? “
> ->We have added another experiment (Section 7.4, table 5) where we hold out specific objects in the training set and test on these unseen objects in the test set. For example, the agent will not see green triangles during training, but will see other green objects and non-green triangles during training and must generalize to the unseen combination at test time. As seen from results in Section 7.4, our method does have a lower completion rate on these tasks but is still able to complete a high completion rate (0.75) and outperform the baselines.
>
> Other improvements: To further improve the experimental comparison, we have also added a number of additional comparisons, comparing to state of the art instruction following methods (Misra 2017, Table 1), pretraining with language (similar to Andreas 2018, Fig 7), using rewards instead of language corrections (Fig 7). We have also provided more analysis regarding the extrapolation and generalization of LGPL in Section 7.4.
>
> [1] Misra, Dipendra Kumar et al. “Mapping Instructions and Visual Observations to Actions with Reinforcement Learning.” EMNLP (2017).
>
> [2] Andreas, Jacob et al. “Learning with Latent Language.” NAACL-HLT (2018).

---

### Official Review · AnonReviewer2 · 2018-11-05
**Interesting problem setup; insufficient experiments**

**Rating:** 6
**Confidence:** 4

**Review:**

This paper provides a meta learning framework that shows how to learn new tasks in an interactive setup.  Each task is learned through a reinforcement learning setup, and then the task is being updated by observing new instructions. They evaluate the proposed method in a simulated setup, in which an agent is moving in a partially-observable environment. They show that the proposed interactive setup achieves better results than when the agent all the instructions are fully observable at the beginning.

The task setup is very interesting. However, the experiments are rather simplistic, and does not evaluate the full capability of the model. Moreover, the current experiments does not convince the reviewer if the claims are true in a more realistic setup. The authors compare the proposed method with one algorithm (their baseline) in which all the instructions are given at the beginning. I am wondering how the method will be compared with a state-of-the-art method that focuses on following instructions, e.g., Artzi and Zettlemoyer work. Moreover, the authors need to compare their method in an environment that has been previously used for other domains with instructions.

---

> ### Author Response · Authors · 2018-11-16
> **R2 Review Response**
>
> Thank you for the detailed and constructive feedback. To address concerns about the experimental setup setup, we have added a new environment that we call robotic object relocation, which involves a continuous state space. Instead of subgoals the corrections here are more relative such as “move a little left”. The results for this environment are in the revised paper in Section 7.3. To address comments about comparisons, we have also added a number of additional comparisons, comparing LGPL to state of the art instruction following methods (Misra 2017, Table 1), pre-training with language (similar to Andreas 2018, Fig 7), using rewards instead of language corrections (Fig 7). To provide a deeper understanding of the methods performance, we have also included a number of additional analyses on extrapolation and generalization in Section 7.4. Please let us know if adding additional comparisons or analysis would be helpful!
>
> We respond to specific comments below:
>
> “I am wondering how the method will be compared with a state-of-the-art method that focuses on following instructions”
> -> We have implemented and compared to state of the art instruction following methods (results in Section 7.2, 7.3) Misra et al. (2017), and pretraining based on language (Andreas et al 2018) which show strong results on instruction following. We find that Misra et al. (2017) performs a little worse than our full information oracle method on the multi-room domain when given all subgoals along with the instruction, and significantly worse when given just the instruction. On the object relocation domain, Misra et al. (2017) performs around the same as our instruction baseline. We would like to emphasize that our work is complementary to better instruction following methods/architectures, it provides us a way to incorporate additional corrections in scenarios where just instructions are misspecified/vague. The specific comparison suggested, Artzi and Zettlemoyer, needs a domain specific executor and a formal language over actions. This approach requires specific engineering for each task and it’s unclear how to create a deterministic executor for ours. We also note that recent state of the art work in instruction following [Andreas 2018], [Misra 17], [Wang 2018], [Janner 2017] do not compare to A+Z for their tasks.
>
> “Moreover, the current experiments does not convince the reviewer if the claims are true in a more realistic setup”
> -> We have now added an additional continuous state space environment, robotic object manipulation, and tested over more varied types of corrections, which demonstrates the applicability of our method to diverse task and correction setups. These results can be found in Section 7.4, and show that our method scales to different setups.
>
> “Moreover the authors need to compare their method in an environment that has been previously used for other domains with instructions”
> -> Our algorithm incorporates language corrections to improve agent behavior quickly on new tasks, when the instruction is vague or ambiguous. No other work to our knowledge studies this problem setting, so we made our own environments for this task - based on existing instruction following domains. Our minigrid environment is a partially observed navigation-based environment and shares structural similarities to existing navigation-based environments such as [Matterport 3D, SAIL, Pond world].
>
> [1] Wang, Xin et al. “Look Before You Leap: Bridging Model-Free and Model-Based Reinforcement Learning for Planned-Ahead Vision-and-Language Navigation.” CoRRabs/1803.07729 (2018)
>
> [2] Andreas, Jacob et al. “Learning with Latent Language.” NAACL-HLT (2018).
>
> [3] Misra, Dipendra Kumar et al. “Mapping Instructions and Visual Observations to Actions with Reinforcement Learning.” EMNLP (2017).

---

### Meta-Review · Area_Chair1 · 2018-12-14
**Innovative interactive instruction setting based on language interaction**

**Confidence:** 5
**Recommendation:** Accept (Poster)

**Metareview:**

The paper proposes a meta-learning approach to "language guided policy learning" where instructions are provided in the form of natural language instructions, rather than in the form of a reward function or through demonstration. A particularly interesting novel feature of the proposed approach is that it can seamlessly incorporate natural language corrections after an initial attempt to solve the task, opening up the direction towards natural instructions through interactive dialogue. The method is empirically shown to be able to learn to navigate environments and manipulate objects more sample efficiently (on test tasks) than approaches without instructions.

The reviewers noted several potential weaknesses: while the problem setting was considered interesting, the empirical validation was seen to be limited. Reviewers noted that only one (simple) domain was studied, and it was unclear if results would hold up in more complex domains. They also note lack of comparison to baselines based on prior work (e.g., pre-training).

The authors provided very detailed replies to the reviewer comments, and added very substantial new experiments, including an entire new domain and newly implemented baselines. Reviewers indicated that they are satisfied with the revisions. The AC reviewed the reviewer suggestions and revisions and notes that the additional experiments significantly improve the contribution of the paper. The resulting consensus is that the paper should be accepted.

The AC would like to note that several figures are very small and unreadable when the paper is printed, e.g., figure 7, and suggests that the authors increase figure size (and font size within figures) to ensure legibility.